

# Ferroptotic cardiomyocyte-derived exosomes promote cardiac macrophage M1 polarization during myocardial infarction

Shengjia Sun[1], Yurong Wu[2], Alimujiang Maimaitijiang[1], Qingyu Huang[1] and Qiying Chen[1]

[1] Department of Cardiology, Huashan Hospital, Fudan University, Shanghai, China
[2] Nursing Department, Huashan Hospital, Fudan University, Shanghai, China

Corresponding author
Qiying Chen,
chenqiying202202@163.com

## ABSTRACT

Ferroptosis is a mode of cell death that occurs in myocardial infarction (MI). Signals emanating from apoptotic cells are able to induce macrophage polarization through exosome-loading cargos, which plays a vital role in the process of disease. However, whether ferroptotic cardiomyocytes derived exosome (MI-Exo) during MI act on macrophage polarization and its mechanism remain unclear. In this study, a MI mouse model was established, and cardiac function evaluation and pathological staining were performed. The effect of MI-Exo on polarization of RAW264.7 cells was assessed by the expression of IL-10 and NOS2. Ferroptosis inhibitor of ferrostatin-1 was used to verify whether MI-Exo function was dependents on ferroptosis. Cardiac function and myocardial histomorphology were markedly impaired and massive immune cell infiltration in MI mice, compared with the sham group. The significantly increased MDA content and $Fe^{2+}$ accumulation in the heart tissue of MI mice suggested cardiomyocyte ferroptosis. Compared with the sham group, the expression of M1 marker NOS2 was significantly up-regulated and M2 marker IL-10 was significantly down-regulated in the heart tissue of MI mice. Exosome-derived from MI HL-1 cell-treated with ferrostatin-1 (Fer-1-Exo) and MI-Exo were internalized by RAW 264.7 cells. Compared with culture alone, co-cultured with MI-Exo significantly promoted NOS2 expression and suppressed IL-10 expression, and decreased proportion of Arginase-1-labeled M2 macrophages, also inhibited phagocytosis of RAW 264.7 cells. Wnt1 and β-cantenin expression also elevated after treated with MI-Exo. However, co-cultured with Fer-1-Exo significantly reversed the above changes on RAW 264.7 cells induced by MI-Exo. In conclusion, ferroptotic cardiomyocytes-derived exosome crosstalk macrophage to induce M1 polarization via Wnt/β-cantenin pathway, resulting in pathological progress in MI. This understanding provides novel therapeutic target for MI.

## INTRODUCTION

Myocardial infarction (MI) is designated as myocardial necrosis due to myocardial ischemia and is a major contributors to mortality of cardiovascular disease in the worldwide (*Nowbar et al., 2019*). Out of the multiple reported forms of cell death during MI process, the five major forms observed in the heart cells are ferroptosis, necroptosis apoptosis, necrosis, and pyroptosis (*Yoshimura et al., 2020*). Ferroptosis is an iron-dependent regulation of cell death, which is concomitant with iron accumulation and lipid peroxidation and discriminate from apoptosis, necrosis and autophagy (*Dixon et al., 2012*). Increasing studies demonstrated that ferroptosis support the MI pathological process, for example, inhibiting ferroptosis by human umbilical cord blood-derived MSCs exosome could attenuate myocardial injury in acute MI mice (*Song et al., 2021*); downregulation of GPX4 triggered ferroptosis in cardiomyocytes which contributed to MI (*Park et al., 2019*); the ferroptosis occurred in acute MI and regulated by Egr-1/miR-15a-5p/GPX4 axis (*Fan et al., 2021*); treatment with Ferrostatin-1, an iron death inhibitor, reduced infarct size and myocardial injury (*Fang et al., 2019*). Therefore, ferroptosis may be a new therapeutic target for MI.

Under non-pathological conditions, macrophages reside in the heart as a non-inflammatory M2 phenotype. After MI occurs, macrophages rapidly recruit to the damaged site to phagocytize and process dead cells to promote tissue repair (*Cheng et al., 2017*). However, recent studies have shown that the recruited macrophages continuously maintain the M1 phenotype, release cytokines and proteases that induce apoptosis in healthy cardiomyocytes resulting in aggravating myocardial damage (*Poon et al., 2014*). Therefore, elucidating the internal factors that induce abnormal activation of macrophages is beneficial to improve the clinical treatment of MI.

Exosomes are extracellular vesicles secreted by cells containing biological information such as proteins and RNA, which are one of the important ways of communication between cells (*Chen et al., 2021*). During MI, exosomes secreted by cardiomyocytes can specifically target macrophages to complete the crosstalk between them (*Loyer et al., 2018*). Previous study focused on exosomes secreted by viable cells or apoptotic cells, while necrotizing cells were rarely investigated. *Dai et al. (2020)* demonstrated that for the first time a link between ferroptotic cells and macrophages, that is, exosomes released by ferroptotic pancreatic cancer cells carry KRAS protein to macrophages, resulting in the M2 polarization of macrophages (*Dai et al., 2020*). Strikingly, cardiomyocyte ferroptosis and macrophage polarization occur simultaneously during MI. Therefore, we asked whether exosomes released by ferroptotic cardiomyocytes are also involved in the regulation of macrophage polarization.

Here, we hypothesized that ferroptotic cardiomyocytes derived exosome crosstalk macrophage to induce M1 polarization, resulting in myocardial damage. In this study, we verified this hypothesis through *in vivo* and *in vitro* experiments, striving to provide new targets for MI treatment.

## MATERIALS AND METHODS

### Mouse model of MI

SPF C57BL/6J male mice were purchased from the Junke Biological Co., LTD. (Nanjing, China). All mice were reared adaptively for one week at 20–24 °C with free access to food and water under natural light with day and night. After that, all mice were randomly divided into sham group ($n = 5$) and MI group ($n = 5$). To construct the MI model, mice were anesthetized with 5% isoflurane inhalation followed by thoracotomy and ligation of the left anterior descending (LAD) coronary artery. For the sham group, mice were underwent only thoracotomy without LAD ligation. At the end of the study, the mice were euthanized by 5% inhalation of isoflurane and cervical dislocation. The heart tissue of mice were isolated and frozen for subsequent experiments. All animal procedures were approved by the Animal Welfare and Ethics Group, Department of Experimental Animal Science, Fudan University (2020 Huashan Hospital JS-574).

### Echocardiographic examination

The left ventricular systolic function of mice was detected by echocardiographic examination. Seven days after the operation, 3 mice in each group were randomly selected to observe their echocardiograms on the Mindray (Mindray Medical Co., Ltd.). Conventional parameters of echocardiography were collected including left ventricular end-diastolic diameter (LVEDD), left ventricular end-systolic diameter (LVESD), fractional shortening (FS), and ejection fraction (EF). EF (%) = (LVEDV-LVESV)/ LVESV $\times100\%$; FS (%) = (LVEDD-LVESD)/LVESD $\times100\%$.

### Hematoxylin-Eosin (H&E) staining

Histopathological changes of the MI-induced heart tissue (3 mice per group) were observed by H&E staining. Seven days after the operation, all mice were euthanized by using 5% isoflurane inhalation followed by cervical dislocation, and then thoracotomy to removed heart tissue. Isolated heart tissue were underwent fixation using 4% paraformaldehyde, and paraffin embedding, and then cut into 5 $\mu$m-thick sections. Later, sections were xylene dewaxed and alcohol gradient dehydration followed by stained with haematoxylin and eosin. Finally, sections were imaged under an optical microscope.

### Ferroptosis indicators measurement

In this study, malondialdehyde (MDA) content and total cellular iron ($Fe^{2+}$) concentration were used to indicate the occurrence of ferroptosis. MDA content and total cellular concentration of $Fe^{2+}$ in heart tissue and HL-1 cells were detected by using a MDA kit (S0131, Beyotime, China) and the colorimetric Iron Colorimetric Assay Kit (#E1042, APPLYGEN, China), respectively, according to the manufacturer's instructions.

### Western blotting

Total protein was extracted using RIPA lysis buffer (Thermo Fisher Scientific, Waltham, MA, USA) from heart tissue and HL-1 cells, and the concentration of the protein was detected by the BCA assay kit (Thermo Fisher Scientific, Waltham, MA, USA). Next, about 20 $\mu$g protein was subjected to 12% SDS-PAGE and then transferred on PVDF

membrane (Sangon, Shanghai, China). The PVDF membranes were washed with water and stained with Ponceau S to confirm transfer, after that blocked with TBST solution contained 5% skimmed milk at 4 °C overnight. Later, PVDF membranes were incubated with primary antibodies at 4 °C overnight followed by incubated with secondary antibody Goat Anti-Mouse IgG H&L(HRP) (1:10000, ab205719, Abcam, Cambridge, UK) at 25 °C for 2 h. The protein bands were imaged by chemiluminescence apparatus (Shanghai Qinxiang, China) and quantified by Image J software. Primary antibodies including: Wnt1 (1:1000, sc514531, Santa Cruz), $\beta$-catenin (1:1000, 8480T, Cell Signaling Technology), IL-10 (1:1000, bs-0698R, Bioss), NOS2 (1:1000, sc7271, Santa Cruz), CD63 (ab68418, Abcam, Cambridge, UK), ACSL4 (1:10000, ab155282, Abcam, Cambridge, UK), TFRC (1:1000, ab214039, Abcam, Cambridge, UK), GAPDH (1:20000, 60004-1-Ig, Proteintech).

## Cell culture and treatment

Mouse cardiac myocytes line of HL-1 was purchased from Shanghai Zhongqiao Xinzhou Biotech (ZQ0920; Shanghai, China) and mouse bone marrow derived macrophages line of RAW 264.7 was purchased from Procell (CP-M141, China). HL-1 cells and RAW 264.7 cells were cultured in DMEM culture medium contained 10% FBS and 1% penicillin/streptomycin (P/S; E607011, Sangon, China) at 37 °C in an atmosphere of 95% air and 5% $CO_2$.

To explore the effect of ferroptosis on exosome derived from cardiac myocytes, HL-1 cells were maintained at hypoxic conditions with 7% $O_2$ and treated with 50 $\mu$M ferrostatin-1 (Fer-1), a ferroptosis inhibitor, for 16 h. HL-1 cells treated with PBS as control. Later, culture supernatants were harvested for exosome isolation.

To explore the role of Wnt/$\beta$-catenin in exosome-mediated macrophage polarization, RAW 264.7 cells were treated with 10 $\mu$M IWR-1 (681669, Sigma-Aldrich), a Wnt/$\beta$-catenin signaling pathway inhibitor, for 48 h. Later, Wnt expression was detected by RT-qPCR and western blot.

## Isolation and identification of exosome

Exosome derived from HL-1 cells was extracted using Exosome Isolation Kit (from cell culture media) (UR52121, Umibio, Shanghai, China) according to the manufacturer's instructions. The morphology and size were observed by transmission electron microscopy (TEM). In briefly, 10 $\mu$L solution of exosome was placed onto the copper wire mesh and precipitate for 1 min. Then, washed with PBS and dropped 10 $\mu$L 2% phosphotungstic acid and precipitate for another 1 min. Finally, whole copper wire mesh was dried at room temperature for 2 min and subjected to image on a TEM (JEM-1200EX, Japan). The concentration and distribution of exosomes were tracked by nanoparticle tracking analysis (NTA) using Zetaview Particle Metrix and its corresponding software ZetaView 8.04.02.

## Uptake of exosomes

The exosome derived from HL-1 cells-treated with erastin or PBS were labeled with DiI-red fluorescent kit (C1036, Beyotime, China), according to the manufacturer's instructions. Next, DiI-labeled exosome were re-suspended in complete medium at a concentration of 5 $\mu$g/mL and then added to a dish for fluorescence microscopy in which RAW 264.7

cells were precultured for 24 h. After 12 h of coculture, the medium was removed by centrifugation and the RAW 264.7 cells were fixed with 4% paraformaldehyde and permeabilized with 0.2% Triton X-100, and stained with DAPI. Finally, RAW 264.7 cells was imaged on a fuorescence microscope. Quantification of exosome uptake was measured by ImageJ software according to mean fluorescent intensity.

## Flow cytometry sorting of macrophages

After incubation of exosomes, M1-polarized and M2-polarized macrophages were separated by flow cytometry. In short, RAW 264.7 cells were centrifuged at 350g for 5 min to remove medium and re-suspended in PBS to incubated with NOS2-Alexa Fluor® 488 (sc-7271, Santa Cruz) and Arginase-1 (D4E3M™) XP® Rabbit (#93668S, Cell Signaling Technology) at 25 °C for 30 min followed by a incubated with Cy3-labeled Goat Anti-Rabbit IgG(H+L) at 25 °C for 30 min in the dark. After centrifugation and resuspension, RAW 264.7 cells were sorted by CytoFLEX LX Flow Cytometer (Beckman, USA).

## Phagocytosis of macrophages

Phagocytosis of the RAW 264.7 cells was investigated using Phagocytosis Assay Kit (Red Zymosan) (ab234054, Abcam, Cambridge, UK), according to the manufacturer's instructions. In briefly, after incubation of exosomes (Macrophages without exosome treatment served as a blank control), RAW 264.7 cells were incubated with 5 μL of prelabeled zymosan particles for another 3 h and then washed with cold phagocytosis assay buffer. Finally, RAW 264.7 cells were observed by fluorescent microscopy. The phagocytic index is equal to the number of zymosan particles in the cell divided by the number of phagocytic cells multiplied by 100%.

## MiRNA mimics and siRNA transfection

MiR-106b-3p mimics and mimics NC were synthesized by Sangon, China. HL-1 cells seed on six-well plates at a density of $2 \times 10^5$ cells/well. When reaching about 80–90% confluence, cells were transfected with miR-106b-3p mimics or mimics NC using Lipofectamine™ 2000 (Invitrogen, Waltham, MA). MiR-106b-3p mimics, mimics NC, and Lipofectamine™ 2000 regent were diluted in 45 μL OPTI-MEM (CORNING). After that, diluted miR-106b-3p mimics was mixed with diluted Lipofectamine™ 2000 regent for 20 min, and then added into HL-1 cells. Cultured for another 24 h and transfection efficiency was measured using RT-qPCR.

## RT-qPCR analysis

Total RNA was isolated using TRIzol reagent (Sigma-Aldrich, St. Louis, MO) and RNA quality was determined by using NanoDrop. To quantify the Wnt1 mRNA expression, RNA was reversed into cDNA using PrimeScript™ RT reagent Kit (Takara, China) and then the qPCR reaction was carried out using SYBR® Premix Ex Taq II (Takara, China) according to the manufacturer's protocol. For detecting the miRNAs expression, RNA was reversed into cDNA using PrimeScript RT reagent kit with gDNA Eraser (Takara, China) followed by qPCR reaction using TB Green Premix Ex Taq (Takara, China). GAPDH was used to normalize mRNA expression, and U6 was used to normalize miRNA expression.

The $2-\Delta\Delta CT$ method was used for calculating the relative expression level. All primers used in this study were shown in Table S1.

## Statistical analysis

The statistical analysis of all data were presented as the mean $\pm$ SD of three independent experiments and processed by software of GraphPad Prism 9.0 (GraphPad Software, USA). Moreover, student's $t$-test was used to compare with two groups and ANOVA followed by Tukey's test was used to compare three groups. $P <0.05$ was considered statistically significant.

# RESULTS

## Cardiomyocyte ferroptosis accompanied by M1 macrophage infiltration during MI

We firstly established a mouse model of MI using LAD ligation and then examined cardiac function with echocardiography. Compared to the sham group, LVIDs ($P = 0.0018$) and LVIDd ($P = 0.00015$) in MI mice were significantly increased (Figs. 1A and 1B). In contrast, the EF ($P = 0.017$) and FS ($P = 0.006$) were substantial reduced post-MI, compared with sham group (Figs. 1A and 1B). Likewise, H&E staining further revealed that histomorphology of myocardial tissues in sham group was normal but badly damaged in MI group, as indicated by disordered cardiomyocyte and massive inflammatory cell infiltration (Fig. 1C). To determine whether myocardial tissue injury is related to ferroptosis, we measured two indicators of ferroptosis such as MDA content and the concentration of $Fe^{2+}$. The results showed that, compared to the sham group, MDA content ($P = 0.049$; Fig. 1D) and the concentration of $Fe^{2+}$ ($P = 0.017$; Fig. 1E) were significantly increased after MI. Moreover, MI induced a significant elevation in the expression of ACSL4 ($P = 0.0067$) and TFRC ($P = 0.0021$), molecular markers of ferroptosis (Fig. 1F), implicating that cardiomyocyte ferroptosis occurred during MI. Next, to explore if the MI altered macrophage polarization in myocardial tissue, we examined the change of polarization marker expression of M1 (NOS2) and M2 (IL-10). As described in Fig. 1G, relative to sham group, NOS2 expression was up-regulated post-MI ($P = 0.0014$), while IL-10 expression was the opposite ($P = 0.0012$), suggesting that MI favor the polarization of M1 macrophages in infarcted area. Taken together, these results suggested that cardiomyocyte ferroptosis accompanied by M1 macrophage infiltration during MI.

## Ferroptotic cardiomyocytes derived exosome identification

A study revealed that ferroptotic pancreatic cancer cells carry KRAS protein to macrophages, resulting in the M2 polarization of macrophages (*Dai et al., 2020*). Our above results proved that cardiomyocyte ferroptosis accompanied by M1 macrophage infiltration in MI, thus, we interrogate that ferroptotic cardiomyocytes derived exosome could crosstalk macrophage to induce M1 polarization. We constructed MI models of HL-1 cells induced by hypoxia, and treated with Fer-1 to mitigate the ferroptosis process. The results showed that Fer-1 treatment significantly alleviated the oxidative stress in hypoxia-induced HL-1 cells, which

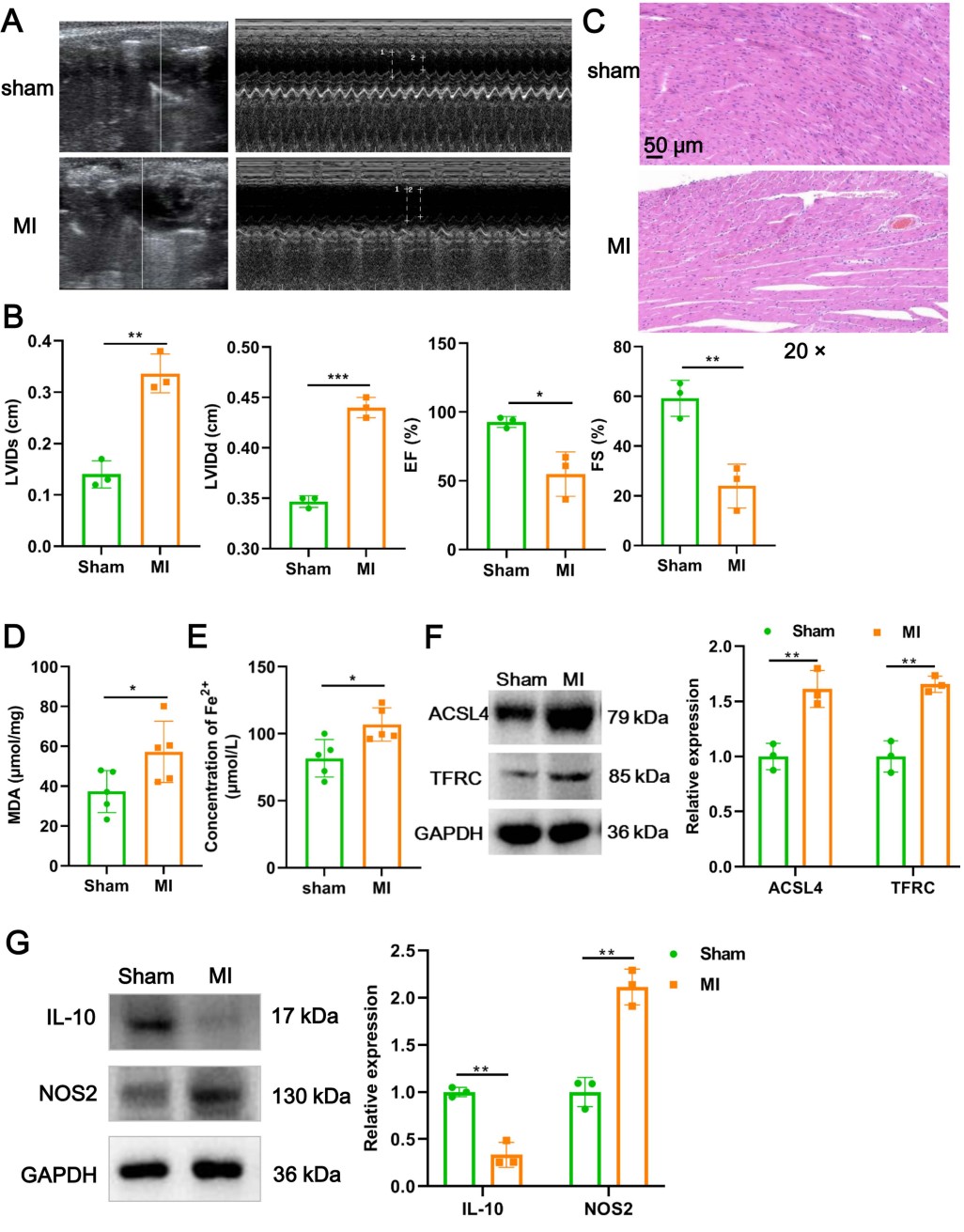

**Figure 1 Cardiomyocyte ferroptosis accompanied by M1 macrophage infiltration during MI.** (A) Representative echocardiographic images of long-axis or short-axis in mice in sham group and MI model group. (B) Quantitative analysis of LVIDs, LVIDd, EF (%), and FS (%) based on the echocardiographic image in mice. (C) Representative images of H&E staining of infarction area in mice. Magnification: 20 ×. (D) MDA content of infarction area in mice. (E) Concentration of $Fe^{2+}$ of infarction area in mice. (F) The relative expression of ferroptosis-related markers of ACSL4 and TFRC of infarction area in mice. (G) The relative expression of IL-10 and NOS2 of infarction area in mice.

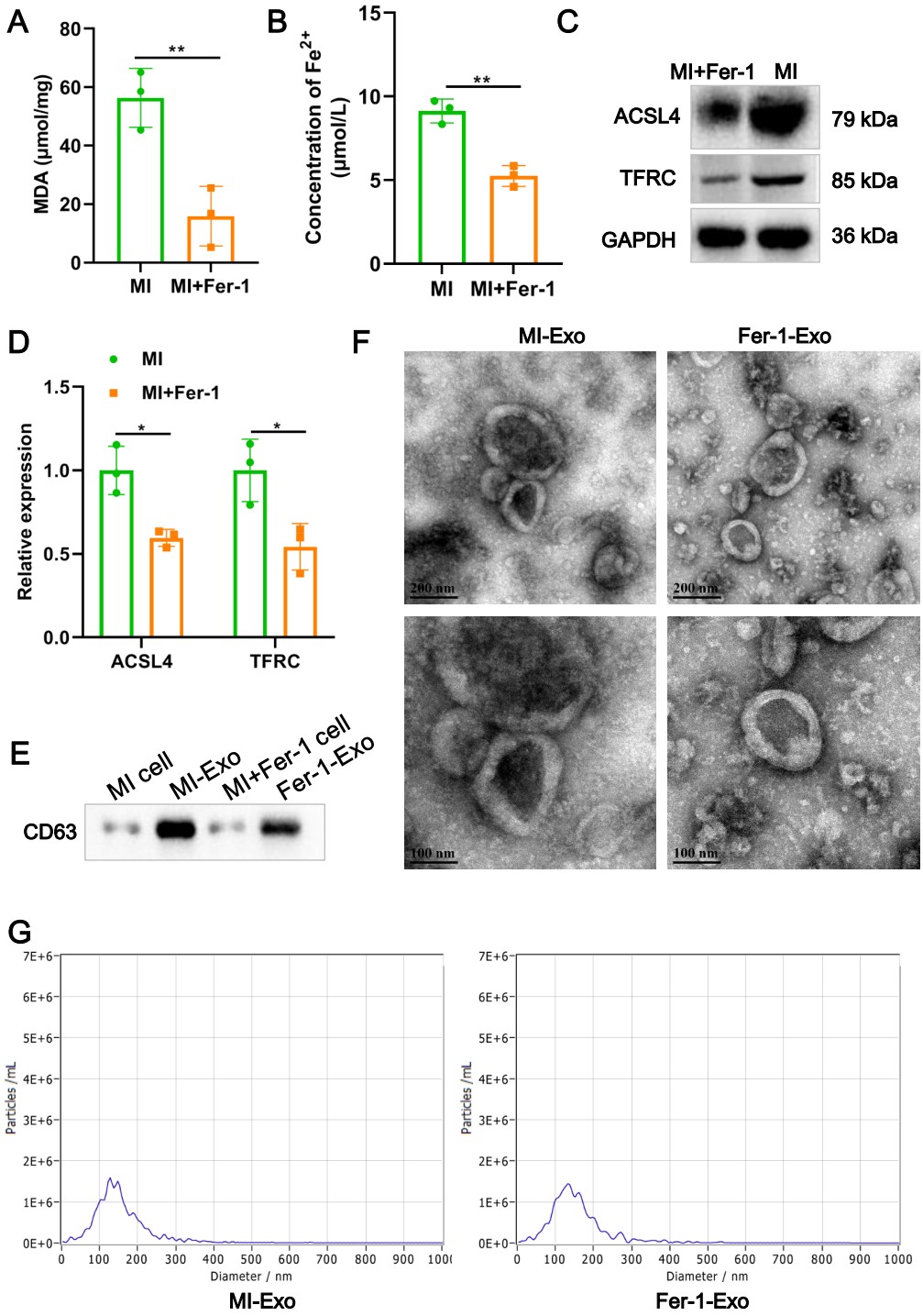

**Figure 2** **Ferroptotic cardiomyocytes derived exosome identification.** (A) MDA content of MI model HL-1 cells treated with Fer-1 or not. (B) Concentration of $Fe^{2+}$ of MI model HL-1 cells treated with Fer-1 or not. (C) The relative protein expression of ferroptosis-related markers of ACSL4 and TFRC in HL-1 cells. (D) Gray statistics of western blot. (E) Exosome marker CD63 was detected by western blot. (F) TEM was used to observe the size and morphology of exosomes. (G) NTA was used to measure size distribution and particle concentration of exosomes.

was manifested by the decrease of MDA content ($P = 0.0081$) and concentration of $Fe^{2+}$ ($P = 0.0022$) (Figs. 2A and 2B). Compared with MI model of HL-1 cells, the protein expression of ACSL4 ($P = 0.010$) and TFRC ($P = 0.027$) also were significantly suppressed by Fer-1 treatment (Figs. 2C and 2D). These results suggested that MI cardiomyocyte undergo ferroptosis and this process can be hindered by Fer-1. Exosomes derived from MI model of ferroptotic HL-1 cells (MI-Exo) and from Fer-1-treated MI model of HL-1 cells (Fer-1-Exo) were isolated and observed by western blot. The expression of exosome marker CD63 was specific enriched in exosome groups relative to cell groups (Fig. 2E). According to TEM, the size of the harvested particles both of MI-Exo and Fer-1-Exo were around 50 nm to 200 nm with an elliptic sphere and central depression (Fig. 2F). NTA revealed that the size distribution of MI-Exo and Fer-1-Exo was $140.3 \pm 66.7$ nm and $140.3 \pm 68.6$ nm, respectively (Fig. 2G). These harvested particles have typical exosome morphology and size indicating that we have successfully isolated MI-Exo and Fer-1-Exo.

## Ferroptotic cardiomyocytes derived exosome promote macrophage M1 polarization

To investigate the effect of ferroptotic cardiomyocytes derived exosome on macrophage M1 polarization, MI-Exo and Fer-1-Exo were used to incubate RAW 264.7 cells. The internalization results of MI-Exo and Fer-1-Exo by RAW 264.7 cells were displayed in Fig. 3A. DiI-labeled exosomes (red fluorescence) were uptaken by RAW 264.7 cells and showed co-localization with DAPI-labeled nuclei (blue fluorescence) in the merge images. The uptake of MI-Exo and Fer-Exo by macrophages was not statistically different (Fig. 3A). The results of western blot presented that compared with blank group, NOS2 expression ($P = 0.013$) was significantly increased in MI-Exo group and IL-10 expression ($P = 0.0199$) was on the opposite (Fig. 3B). This expression trend was consistent with the results of animal experiments. Furthermore, compared with MI-Exo group, Fer-1 treatment decreased NOS2 expression ($P = 0.015$) but elevated IL-10 expression ($P = 0.035$) in hypoxia-induced HL-1 cells. Arginase-1 antibody-labeled flow cytometry detected the proportion of M2 phenotype macrophages in each treatment group. Flow cytometry results also supported that MI-Exo caused a decrease ($P = 0.046$) in proportion of M2 phenotype macrophage and conversely, Fer-1 treatment promoted ($P = 0.014$) the proportion of M2 phenotype differentiation (Fig. 3C). Therefore, these results suggest that ferroptotic cardiomyocyte-derived exosomes promote macrophage M1 phenotype differentiation during MI and that ferroptosis inhibitor treatment favors macrophage differentiation to M2 phenotype.

## Ferroptotic cardiomyocytes derived exosome repaired phagocytosis of RAW 264.7 cells

Macrophage polarization leads to functional alterations in their behavior. Next, we explored whether MI-Exo induced macrophage M1 polarization could change functional phagocytosis. Phagocytosis of zymosan particles by macrophage existed in all three group (Fig. 4A), and then phagocytic cells and phagocytic index were photometrically quantified based on the fluorescently labeled zymosan uptake. As shown in Fig. 4B, there was no difference in the number of phagocytic macrophages between blank group and MI-Exo group ($P = 0.24$), but significantly reduced in Fer-1-Exo group ($P = 0.045$). However,

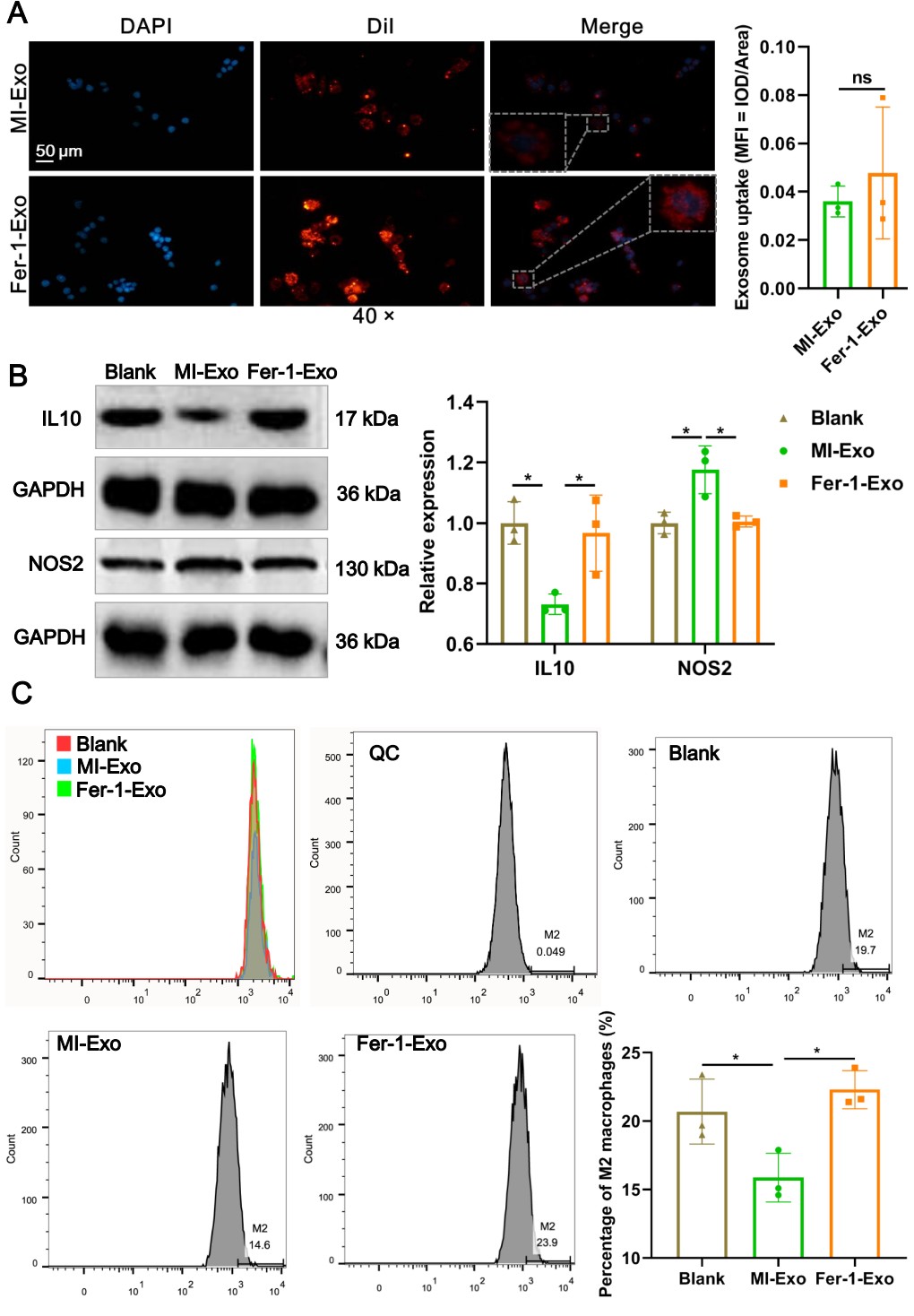

**Figure 3** **Ferroptotic cardiomyocytes derived exosome promote macrophage M1 polarization.** (A) DiI-labeled exosomes were internalized by macrophages. Magnification: 40×. (B) The relative expression of IL10 and NOS2 in RAW 264.7 macrophages cultured alone (Blank group), co-cultured with MI model HL-1 cells-derived exosomes (MI-Exo group), or co-cultured with exosomes derived from MI model HL-1 cells-treated with Fer-1 (Fer-1-Exo group). (C) The proportion of M2 phenotype macrophage detected by Arginase-1 antibody-labeled flow cytometry in Blank group, MI-Exo group, and Fer-1-Exo group.

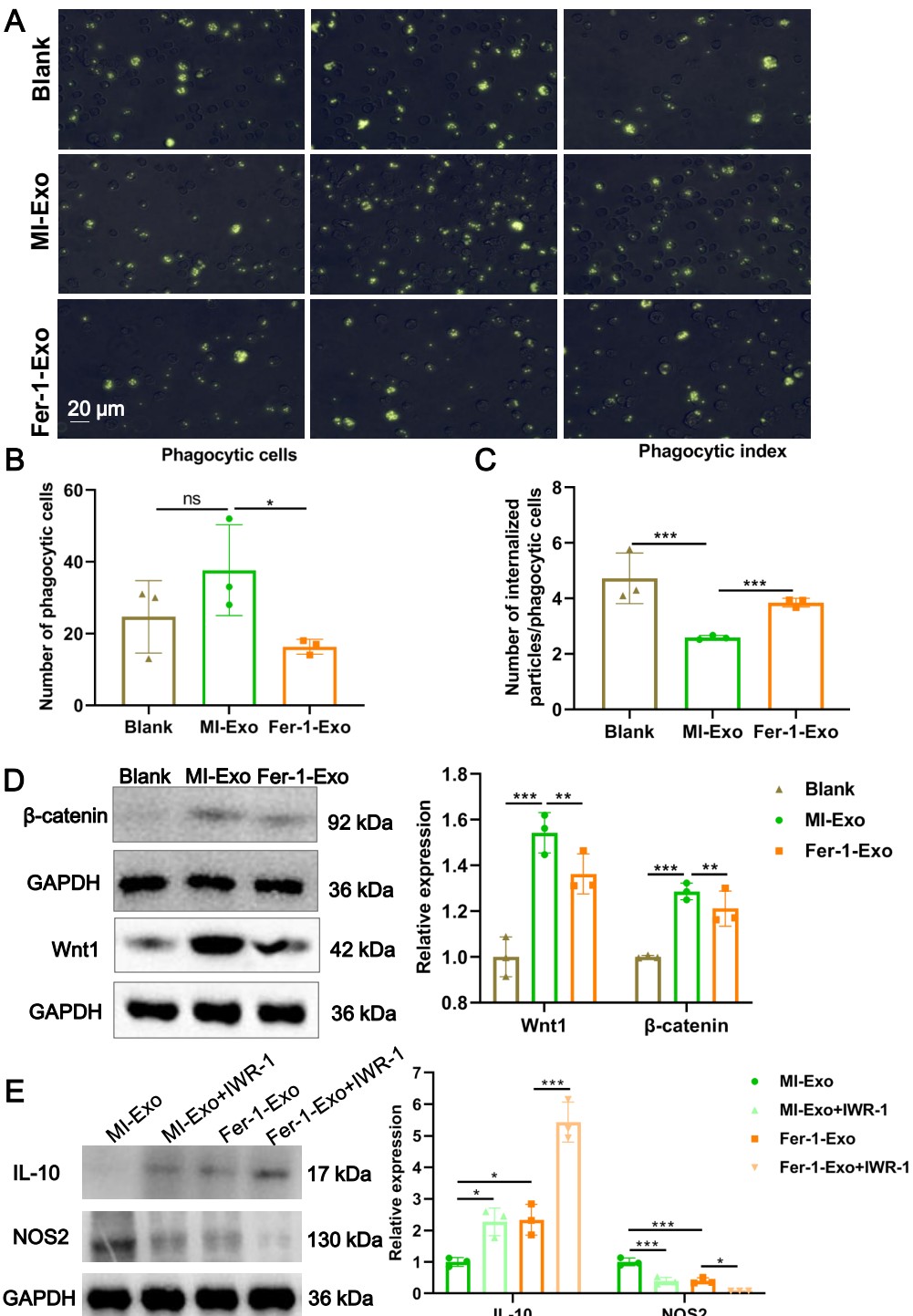

**Figure 4** **Ferroptotic cardiomyocytes derived exosome repaired phagocytosis of RAW 264.7 cells.** (A) Images of macrophage capacity of phagocytize fluorescent-labeled zymosan particles. (B) The number of phagocytic macrophages in Blank group, MI-Exo group, and Fer-1-Exo group. (C) The phagocytic index of macrophage in Blank group, MI-Exo group, and Fer-1-Exo group. Phagocytic index = number of internalized particles/phagocytic cells. (D) The relative expression of Wnt1 and β-catenin in of macrophages in Blank group, MI-Exo group, and Fer-1-Exo group. (E) The relative expression of IL-10 and NOS2 in exosome pretreated-macrophages administrated with IWR-1, a Wnt singling pathway inhibitor.

phagocytic index of macrophage was significantly down-regulated after co-cultured with MI-Exo ($P < 0.0001$), and then revised by Fer-1 treatment ($P = 0.0002$; Fig. 4C), suggesting that MI-Exo repaired phagocytosis of macrophage but this repairment can be partially eliminated by ferroptosis inhibitor.

It has been reported that the Wnt signaling pathway is responsible for macrophage polarization (*Abaricia et al., 2020*). Therefore, we detected classical molecules of the Wnt signaling pathway, Wnt1 and β-catenin, to clarify the activation of RAW 264.7 cells polarization signals during MI. As expected, compared with blank group, the expression of Wnt1 ($P = 0.0007$) and β-catenin ($P = 0.001$) were up-regulated in MI-Exo group, suggesting that Wnt signaling pathway was activated and involved in macrophage M1 polarization during MI progression, whereas this activation of Wnt signaling pathway was suppressed by Fer-1 treatment (Wnt1: $P = 0.0054$; β-catenin: $P = 0.0048$; Fig. 4D). To further confirm above results, the specific Wnt signaling pathway inhibitor IWR-1 was applied. As shown in Fig. 4E, MI-Exo-induced expression of M1 polarization marker NOS2 was significantly inhibited by IWR-1 (MI-Exo *vs* MI-Exo+ IWR-1, $P = 0.0003$), and the synergy of IWR-1 and Fer-1 could further reduce the expression of NOS2 (Fer-1-Exo *vs* Fer-1-Exo+IWR-1, $P = 0.016$). The effect of IWR-1 on the M2 polarization marker IL-10 was completely opposite (Fig. 4E). These results indicated that the Wnt signaling pathway is involved in ferroptotic cardiomyocytes derived exosome induced macrophage polarization during MI.

## Ferroptotic cardiomyocytes derived exosome promote macrophage M1 polarization *via* miR-106b-3p

We further explored which signal moleculars mainly carried by ferroptotic cardiomyocytes derived exosome to regulate Wnt signaling pathway and ultimately affect macrophage polarization. We first analyzed the mRNAs that can be encapsulated in exosomes through the ExoCarta database (http://exocarta.org/download) and found that there are only 5 exosomal mRNAs (Actb, Pdcd6ip, C3, RAB14, and F5) in common between humans and mice, but these mRNAs were not associated with the Wnt signaling pathway. Existing studies point towards exosomes may affect macrophage polarization by delivering miRNAs (*Ma et al., 2022*), these reminded us to pay attention to the change in miRNA profile in MI-Exo. As shown in Fig. 5A, there were 54 overlaps between human exosome and mouse exosome. In addition, the results of miRWalk database (http://mirwalk.umm.uni-heidelberg.de/) analysis showed that 148 miRNAs that can targeted regulate Wnt1 are shared by human and mice (Fig. 5B). Next, a total of 21 potential miRNAs were obtained from the intersection of the aforementioned two overlapping results (Fig. 5C). The presence of these 21 miRNAs in MI-Exo and Fer-1-Exo was verified thereafter using RT-qPCR. A heatmap displayed the expression results of 21 miRNAs and revealed that only 7 miRNAs were significantly up-regulated in Fer-1-Exo compared with MI-Exo, of which miR-106b-3p had the highest fold up-regulation ($P = 0.00092$; Fig. 5D). Thus, we focused on miR-106b-3p. We subsequently examined the miR-106b-3p expression on macrophages after incubation with MI-Exo or Fer-1-Exo. As expected, Fer-1-Exo incubated-macrophages had higher miR-106b-3p expression than MI-Exo group ($P < 0.0001$; Fig. 5E). We thereafter determined

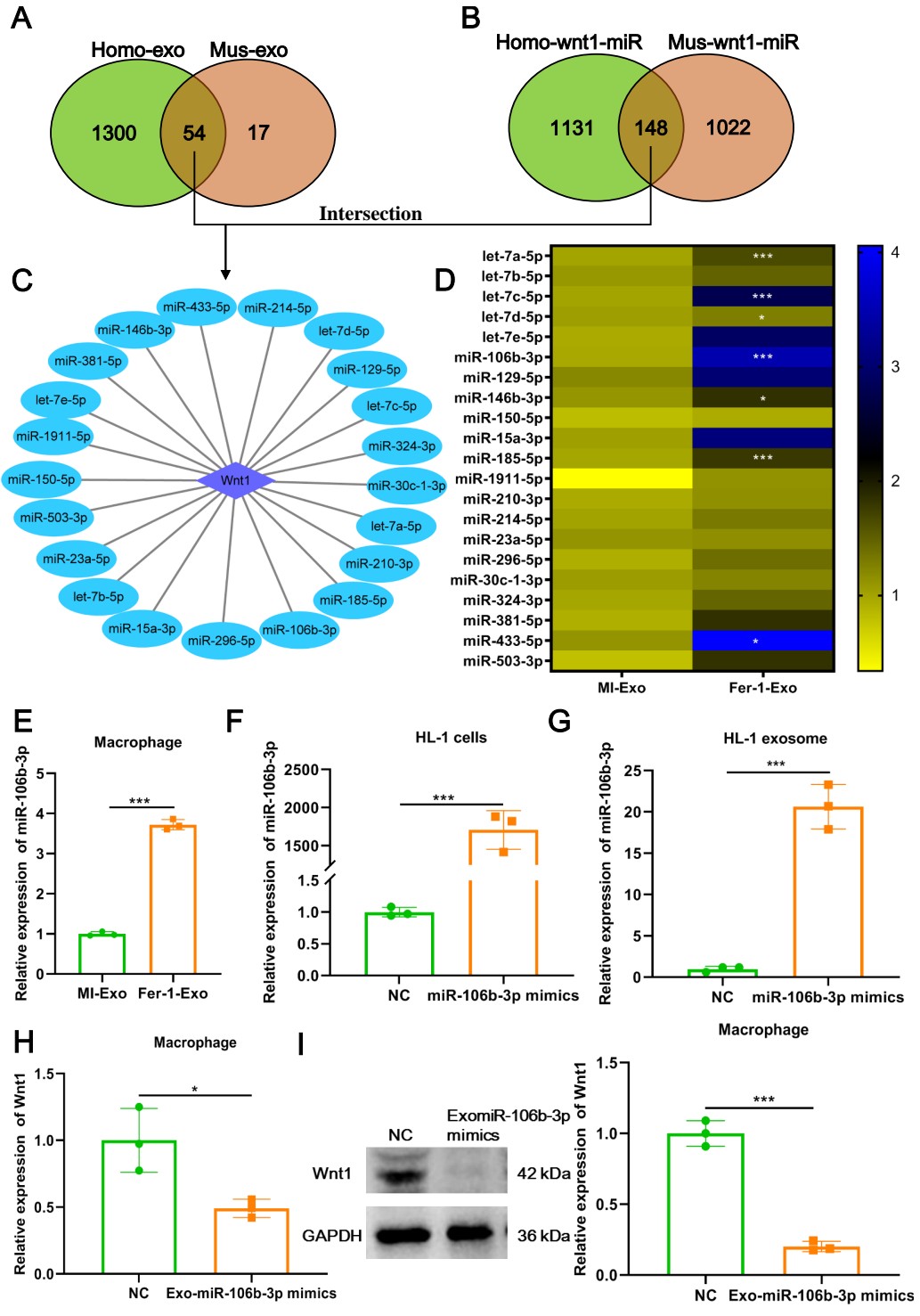

**Figure 5** **Ferroptotic cardiomyocytes derived exosome promote macrophage M1 polarization via miR-106b-3p.** (A) MiRNA profile exists in human exosome or mouse exosome, and there were 54 miRNAs overlap between them. (B) A total of 148 miRNAs that can targeted regulate Wnt1 are shared by humans and mice. (C) Network of Wnt1-miRNAs. A total of 21 potential miRNAs were obtained from the intersection of the aforementioned two overlapping results. 

**Figure 5 (...continued)**
(D) The heatmap of expression of 21 potential miRNAs in MI-Exo and Fer-1-Exo using RT-qPCR. (E) The expression of miR-106b-3p on macrophages after incubation with MI-Exo or Fer-1-Exo. (F) The overexpression efficiency of miR-106b-3p mimics in HL-1 cells detected by RT-qPCR. (G) The overexpression efficiency of miR-106b-3p mimics in HL-1 cell-derived exosome detected by RT-qPCR. (H) The mRNA expression of Wnt1 on macrophages after incubation with NC or Exo-miR-106b-3p mimics. (I) The protein expression of Wnt1 on macrophages after incubation with NC or Exo-miR-106b-3p mimics. * means $P < 0.05$, *** means $P < 0.001$.

whether the miR-106b-3p carried by MI-Exo regulates macrophage polarization by Wnt1. The overexpression efficiency of miR-106b-3p mimics was good both in HL-1 cells ($P = 0.00031$; Fig. 5F) and HL-1 cells-derived exosome ($P = 0.00023$; Fig. 5G). Then, the exosomes overexpressing miR-106b-3p were co-cultured with macrophages. The results showed that Exo-miR-106b-3p mimics significantly suppressed the Wnt1 expression at mRNA ($P = 0.024$; Fig. 5H) and protein level ($P = 0.00014$; Fig. 5I), suggesting that miR-106b-3p targeted regulate Wnt1 in macrophages. Collectively, in MI model, ferroptosis leads to a decrease in the content of exosomal miR-106b-3p released by cardiomyocytes, resulting in the activation of Wnt1, which ultimately promotes M1 polarization in macrophages, whereas Fer-1 can block ferroptosis and restore the content of exosomal miR-106b-3p in cardiomyocyte thereby Fer-1 has a therapeutic effect.

## DISCUSSION

MI occurs when the coronary artery that supply the cardiomyocytes with nutrients and oxygen is blocked. After MI, cardiomyocytes undergo various forms of death including ferroptosis (*Del Re et al., 2019*). Under normal conditions, cardiac macrophages rapidly engulf and process these dead cells to repair damaged heart tissue (*Yoshimura et al., 2020*). However, studies have found that cardiac macrophages have persistent activation of the M1 phenotype, and abnormally activated M1 macrophages become the culprits of the inflammatory amplification cascade, which ultimately leads to deterioration of cardiac function (*Liu et al., 2020*; *Yoshimura et al., 2020*). In this study, we found that exosome secreted by ferroptotic cardiomyocytes was responsible for promoting macrophage M1 polarization during MI *in vivo* and vitro. We also found that pretreated with ferroptosis inhibitor to reduce HL-1 ferroptosis can attenuate the promotion of macrophage M1 polarization by exosome derived from MI model HL-1 cells.

Signals from ferroptosis-dead cells can affect the function of recipient cells including macrophages. *Caruso & Poon (2018)* reviewed that apoptotic cell-derived extracellular vesicles (EVs) crosstalk immune cells to aid removal of dying cells *via* releasing intercellular signals of "Find-Me" and "Eat-me", and modulating antigen presentation *via* diverse mechanism (*Caruso & Poon, 2018*). Her review made us realize that crosstalk between dead cells and immune cells is also important for disease progression. However, very little research has been done on the effects of signals from ferroptotic cells on immune cells. *Dai et al. (2020)* demonstrated for the first time a link between ferroptotic cells and macrophages, that is, exosomes released by ferroptotic pancreatic cancer cells carry KRAS protein to macrophages, resulting in the M2 polarization of macrophages. *Ito et al.*

*(2021)* first evidenced that ferroptosis-dependent EVs from macrophage by loading ferritin contribute to mesothelial carcinogenesis (*Ito et al., 2021*). These results led us to wonder whether ferroptotic cardiomyocytes acted on macrophages function change through exosomes. Excitingly, similar to the above results, ferroptosis-dependent exosomes from cardiomyocytes significantly promoted macrophage M1 polarization in MI model.

Wnt signaling pathway is an evolutionarily conserved pathway, an important molecular signal that is indispensable for maintaining life activities, and is involved in a variety of important cellular processes, such as cellular polarity, proliferation and migration (*Loh, Van Amerongen & Nusse, 2016*). Importantly, macrophages accumulation and polarization also modulated by Wnt signaling pathway (*Cosin-Roger, Ortiz-Masia & Barrachina, 2019*). For example, Wnt/β-catenin signaling activation by IL-17A induced RAW264.7 macrophage M1 polarization and inhibited M2 polarization (*Yuan et al., 2020*). On the contrary, Wnt/β-catenin signaling activation by Dickkopf-2 knockdown dramatically inhibited expression of M1 polarized macrophage markers but promoted M2 (*Zhang et al., 2021*). Wnt/β-catenin signaling activation by Wnt3a treatment exacerbated IL-4- or TGF1-induced macrophage M2 polarization resulting in kidney fibrosis (*Feng et al., 2018*). Those studies indicated a dual pro- and anti- polarization role of Wnt/β-catenin signaling regarding cellular context. However, the role of Wnt/β-catenin signaling activation inducd macrophage polarization under MI pathological condition has not been reported. In this study, we found that a consequence of Wnt/β-catenin signaling activation is macrophage M1 polarization, which contributes to MI progression.

There are some limitations in this study regretfully. One is lack of confirmatory testing for clinical sample and the other did not detect the expression of markers that in the ferroptosis cascade. The third limitation is that the contents of exosomes are not detected, so it is still unclear what the effective signal to macrophages is.

## CONCLUSIONS

In summary, *in vitro* and *in vivo* MI models have shown that cardiomyocytes undergo ferroptosis. Ferroptotic cardiomyocyte-derived exosomes act on cardiac macrophages to skew toward an M1-polarized phenotype by activating the Wnt/β-catenin signaling pathway. Exogenous pharmacological inhibition of ferroptosis, then Fer-1-Exo can inhibit Wnt/β-catenin signaling pathway activation and M1 polarization. The present study provides basic insight of crosstalk between ferroptotic cardiomyocyte and macrophage for MI and provides a treatment target for MI.

### Funding

The authors received no funding for this work.

### Competing Interests

The authors declare there are no competing interests.

## Author Contributions

- Shengjia Sun conceived and designed the experiments, performed the experiments, prepared figures and/or tables, authored or reviewed drafts of the article, and approved the final draft.
- Yurong Wu analyzed the data, prepared figures and/or tables, and approved the final draft.
- Alimujiang Maimaitijiang analyzed the data, authored or reviewed drafts of the article, and approved the final draft.
- Qingyu Huang performed the experiments, prepared figures and/or tables, and approved the final draft.
- Qiying Chen conceived and designed the experiments, authored or reviewed drafts of the article, and approved the final draft.

## Animal Ethics

The following information was supplied relating to ethical approvals (i.e., approving body and any reference numbers):

Animal Welfare and Ethics Group, Department of Experimental Animal Science, Fudan University (2020 Huashan Hospital JS-574).

## Data Availability

The raw data is available in the Supplemental File.

## Supplemental Information

Supplemental information for this article can be found online at http://dx.doi.org/10.7717/peerj.13717#supplemental-information.

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
