# Peer review of "Ferroptotic cardiomyocyte-derived exosomes promote cardiac macrophage M1 polarization during myocardial infarction"

_PeerJ, doi:10.7717/peerj.13717_

## Round 0.1 · original submission · Minor Revisions

Please address the concerns of all reviewers and revise the manuscript accordingly.

Reviewer 1 ·

Basic reporting

In the manuscript titled ‘Ferroptotic cardiomyocyte-derived exosomes promote cardiac macrophage M1 polarization during myocardial infarction’, the authors investigate the possible role of ferroptotic cardiomyocytes derived exosome on macrophage polarization in myocardial infarction (MI). Overall, the experiments are well controlled and well-reported, results are clearly presented, and the manuscript is well written. Although the authors imply that they investigate the role and mechanism through which exosomes may induce macrophage polarization (abstract lines 18-19), more experiments are needed to fully delineate this scientific phenomenon.

- One of the hallmarks of ferroptosis is lipid peroxidation. Although authors measure malondialdehyde content and total cellular iron concentration, from a mechanistic perspective, these analyses are insufficient. It is recommended that authors look at additional molecular markers/readouts (I do not expect authors to perform all of the following, but a couple of these analyses are highly recommended)
- 1. LC-MS/MS analysis to analyze changes in lipid profile
- 2. Using lipid-soluble fluorescent probes
- 3. Looking at the molecular markers of ferroptosis (ACSL4 for instance)


One of the major drawbacks of this study is the lack of characterizing exosomes. Without identifying the key ‘signal’, the claim of exosome-mediated regulation of macrophage polarization is weak. Although it might be challenging to conclusively determine and prove the signaling molecule, getting pretty close to a few candidates by determining exosome content and defining changing exosome composition following ferroptosis is possible (and needed).

Experimental design

Generally, the experiments are well controlled, well-reported and the methodologies are described with appropriate details.

However, as suggested above, the determination of additional ferroptosis markers is needed and importantly, experiments directed at determining the key signaling molecules in exosomes are needed. At the very least, determining exosome contents and analyzing changes in exosome contest following ferroptosis is necessary.

Validity of the findings

Authors probe an important scientific idea that has a huge potential in the development of novel diagnostic and/or therapeutic targets for myocardial infarction. However, as a step closer to that goal and to complete the scientific analyses, further work is needed.

Additional comments

Overall, nicely written manuscript and an interesting scientific study.

·

Basic reporting

The manuscript by Sun et.al and colleagues are exploring the role of ferroptosis cardiomyocytes derived exosomes in M1 macrophage polarization. The authors provide compelling evidence of M1 macrophage infiltration upon cardiomyocyte ferroptosis and ferroptosis derived exosomes promote M1 macrophage polarization. Overall, the data support the conclusions very well and I have only a few suggestions for improving the manuscript.

Experimental design

No comments

Validity of the findings

1. Figure 1C, 3A, and Fig 4A: Scale bars are missing.

2. Authors quantified the ferroptosis indicator MDA and Fe+2 concentration but it looks like there is approximately a 20% difference between the Sham group and MI group. Can the author comment on what Fe+2 concentration could induce ferroptosis?

3. Although the authors used ferroptosis indicators such as MDA and Fe+2 concentration, the Authors can try ferroptosis biomarker transferrin receptor TFRC and check whether there is any difference between the shame and MI group.

4. Figure 1F: It’s difficult to evaluate the western blot data. I don’t any expression of IL-10 in the MI group and it’s hard to see any difference in NOS2 levels. Authors could have tried using Immunohistochemistry with a similar maker or better marker such as CD86 or CD206 which can differentiate M1 and M2.

5. How does the author characterize the exosomes other than the particle size quantification? Also, how to distinguish exosomes derived from the ferroptosis cardiomyocytes from other exosomes?

6. Fig 3A: Authors could have discussed a bit more about exosomes uptake by RAW 264.7 cells. Is it possible to quantify?

7. Fig 4A: How do authors identify or distinguish that these are M1 macrophages?

·

Basic reporting

This work is very well designed, executed, and discussed.

Experimental design

Shengjia Sun and colleagues are trying to identify the reason behind M1 macrophage polarization in myocardial infarction and trying to establish if this is exerted by exosomes derived from ferroptotic cardiomyocytes (MI-Exo). The authors have succeeded in providing compelling evidence for the involvement of exosomes derived from ferroptotic cardiomyocytes in M1 macrophage polarization. The underlying mechanism was also identified as Wnt/ ß-catenin pathway activation. In addition, they also show that inhibition of Ferroptosis by Ferrostatin-1 results in abrogating the effect of exosomes. Overall, the data support the conclusions very well and I have only a few suggestions for improving the manuscript.

1) The authors have indicated that RAW 264.7 cells uptake DiI-labelled exosomes, but it is not clear by which pathway they are engulfed by macrophages, and inhibiting their uptake affects the resultant effect of exosomes in M1 macrophage polarization?
2) The authors have indicated the uptake of both MI-Exo & Fer-1-Exo by Raw 264.7 cells; however, it seems that uptake of Fer-1-Exo is higher than MI-Exo as shown in Fig. 3A. It would be interesting to know if this is due to the differential labeling of respective exosomes or their differential uptake. And if it is differential uptake, what is the reason behind this, and does it have any effect on subsequent experiments?
3) In figure-4A, where authors are comparing phagocytosis in the blank, MI-Exo & Fer-1-Exo, it seems there are fewer cells in Fer-1-Exo than in Blank & MI-Exo conditions. Are Fer-1-Exo toxic to macrophages?

Validity of the findings

4)The Wnt/ß-catenin pathway involvement should be confirmed by another experiment such as Wnt signaling inhibition and its effect on macrophage polarization.

Additional comments

A few minor points.

5) the scale bar is missing in Fig.-3A & 4A.
6) What is the unit of Fe2+ concentration in Fig.1E & 2 B. This is for the readers to understand Fe2+ levels in vitro and in vivo under physiological conditions?
7)Presentation of flow data can be improved by providing staggered overlay histogram and inclusion of unstained control etc.

---

## Round 0.2 · Minor Revisions

Please address the remaining concerns of reviewer #3.

Reviewer 1 ·

Basic reporting

The authors have satisfactorily addressed the concerns raised by reviewers. I have no objection to recommending this manuscript for publication in PeerJ.

Experimental design

Authors have addressed concerns raised by reviewers by supplementing their work with additional experiments such as analysis of molecular markers of ferroptosis.

Validity of the findings

Please see above.

Additional comments

No additional comments.

·

Basic reporting

The authors have now answered all of my requests and suggestions.

Experimental design

No comment

Validity of the findings

No comment

Additional comments

I am now happy for this manuscript to be published.

·

Basic reporting

The paper has been revised accordingly.

Experimental design

The authors have revised the manuscript with relevant experiments.

Validity of the findings

Everything looks great.

Additional comments

The flow data needs a very very minor correction. In Fig.3C, in the overlay histogram, instead of overlaying the mCherry Fluorescence channel, authors have overlaid either Forward Scatter/Side-Scatter. a very minor correction is needed.

---

## Round 0.3 · accepted · Accept

The remaining concerns of the reviewer were addressed and the revised manuscript is acceptable now.